# Carneusones A-F, Benzophenone Derivatives from Sponge-Derived Fungus *Aspergillus carneus* GXIMD00543

**DOI:** 10.3390/md22020063

**Published:** 2024-01-25

**Authors:** Chun-Ju Lu, Li-Fen Liang, Geng-Si Zhang, Hai-Yan Li, Chun-Qing Fu, Qin Yu, Dong-Mei Zhou, Zhi-Wei Su, Kai Liu, Cheng-Hai Gao, Xin-Ya Xu, Yong-Hong Liu

**Affiliations:** Guangxi Key Laboratory of Marine Drugs, Institute of Marine Drugs, Guangxi University of Chinese Medicine, Nanning 530200, China; luchunjv@163.com (C.-J.L.); 15277833592@163.com (L.-F.L.); 15534445495@163.com (G.-S.Z.); lihaiyan12368@163.com (H.-Y.L.); 18378464907@163.com (C.-Q.F.); 19862354152@163.com (Q.Y.); zhoudm@gxtcmu.edu.cn (D.-M.Z.); suzw1454@126.com (Z.-W.S.); liuk@gxtcmu.edu.cn (K.L.); gaoch@gxtcmu.edu.cn (C.-H.G.)

**Keywords:** marine fungus, anti-inflammatory activity, *Aspergillus carneus*, benzophenone, Beibu gulf

## Abstract

Six benzophenone derivatives, carneusones A-F (**1**–**6**), along with seven known compounds (**7**–**13**) were isolated from a strain of sponge-derived marine fungus *Aspergillus carneus* GXIMD00543. Their chemical structures were elucidated by detailed spectroscopic data and quantum chemical calculations. Compounds **5**, **6**, and **8** exhibited moderate anti-inflammatory activity on NO secretion using lipopolysaccharide (LPS)-induced RAW 264.7 cells with EC_50_ values of 34.6 ± 0.9, 20.2 ± 1.8, and 26.8 ± 1.7 μM, while **11** showed potent effect with an EC_50_ value of 2.9 ± 0.1 μM.

## 1. Introduction

Marine microorganisms, including fungi, are widely distributed in various marine environments, such as the deep sea, polar regions, even in the body of the host organisms including marine animals and plants [1]. As a result of their long-term existence in extreme survival environments, marine fungi have evolved unique metabolic methods and could produce abundant structurally diverse secondary metabolites [2,3]. Compounds isolated from marine fungi have also attracted considerable attention for their broad range of biological activities and have become important sources of novel drugs [4]. In the last five years, over one third of bioactive natural marine products were obtained from marine fungi and mangrove fungi [5].

Natural benzophenone derivatives, or dipenyl ketone analogues, are a class of compounds widely distributed in nature with a phenol–carbonyl–phenol skeleton. More than 300 benzophenone derivatives have been isolated from plants and fungi. They exhibit great structural diversity, which is attributed to protons replaced by hydroxyl, alkyl, alkyloxy groups, or halogen atoms [6,7]. Owing to their variable substituents, many benzophenone derivatives show a range of biological activities including anticancer, anti-inflammatory, antimicrobial, and antiviral effects [8].

*Aspergillus* is an important fungal genera producing benzophenones derivatives. Four xanthones oxisterigmatocystins A-C, and 5-methoxysterigmatocystin were isolated from one strain of deep-sea-derived fungus *Aspergillus versicolor*. 5-Methoxysterigmatocystin exhibited moderate cytotoxicities against the A-549 and HL-60 cell lines with IC_50_ values of 3.86 and 5.32 μM [9]. New benzophenone derivatives versixanthones A-F were retrieved from a mangrove endophyte *Aspergillus versicolor* HDN1009. They exhibited cytotoxicities against seven cancer cell lines (HL-60, K562, A549, H1945, 803, HO-8910, and HCT-116) with IC_50_ values ranging from 0.7 to 20.8 μM. Compound versixanthone F showed potent cytotoxicity against HCT-116 cells with an IC_50_ value of 0.7 μM [10]. Compounds monochlorsulochrin, dihydrogeodin, methyl-(2-chloro-l,6-dihydroxy-3-methylxanthone)-8-carboxylate, and methyl-(4- chloro-l,6-dihydroxy-3-methylxanthone)-8-carboxylate) were isolated from the extract of a strain of *Aspergillus flavipes* DL-11. They exhibited antibacterial activities against six pathogenic Gram-positive bacteria *Staphylococcus aureus* (ATCC43300), *Staphylococcus aureus* (ATCC29213), *Staphylococcus aureus* (ATCC33591), *Staphylococcus aureus* (ATCC25923), *Enterococcus faecalis* (ATCC51299), and *Enterococcus faecium* (ATCC35667) with MIC values ranging from 1.56 to 50 μg/mL [11]. 

As part of our ongoing search for bioactive compounds from marine fungi, *Aspergillus carneus* GXIMD00543, which is associated with sponge sample obtained from Weizhou Island, Beibu Gulf, was selected for further studies. Chemical investigation of the fungal extract led to the isolation of six benzophenone derivatives (**1**–**6**), along with seven known compounds (**7**–**13**) (Figure 1). The anti-inflammatory potential of the compounds was assayed. Herein, we reported the details of the isolation, structural elucidation, and determination of the anti-inflammatory effect of those compounds.

## 2. Results and Discussion

### 2.1. Strain Isolation and Species Identification

The fungus was identified as *Aspergillus carneus* based on its morphological features and the sequence analysis of the ITS region of the rRNA gene (Appendix A), which exhibited 99.81% similarity with fungal strain *A. carneus*, with GenBank accession number MH777426.1. 

### 2.2. Elucidation of Chemical Structures

Compound **1** was obtained as yellowish powder. The molecular formula was determined to be C_15_H_10_O_7_ according to the HRESIMS peaks at *m*/*z* 303.0510 ([M + H]^+^, calcd. 303.0499) and 325.0330 ([M + Na]^+^, calcd. 325.0319). The ^1^H NMR spectrum of **1** exhibited signals of three aromatic protons at δ_H_ 6.81 (1H, s, H-2), 6.87 (1H, s, H-5), and 6.61 (1H, s, H-7), one methyl proton at δ_H_ 2.39 (3H, s, H-11), and one hydroxyl group signal at the low field region δ_H_ 12.57 (1H, br s, OH-8), which formed an intramolecular hydrogen bond with the carbonyl group at C-9 (Table 1). The ^13^C NMR and HSQC spectra of **1** exposed the presence of one methyl group at δ_C_ 22.0 (C-11), three aromatic methines at δ_C_ 107.3 (C-5), 110.9 (C-7), and 111.9 (C-2), eleven quaternary carbons, including nine aromatic carbons, at δ_C_ 105.6 (C-8a), 109.1 (C-9a), 126.6 (C-1), 133.3 (C-4), 145.9 (C-3), 148.4 (C-6), 152.3 (C-4a), 155.3 (C-4b), and 160.7 (C-8), and two carbonyl carbons at δ_C_ 170.1 (C-10) and 179.8 (C-9) (Table 2). The HMBC spectrum showed correlations from H-2 to C-1, C-3, C-4, C-9a, and C-10, from H-5 to C-4b, C-7, C-8a, C-9, and C-11, from H-7 to C-5, C-8, C-9, and C-11, and from CH_3_-11 to C-5, C-6, and C-7 (Figure 2). All these data exhibited close similarity with compound 3-hydroxy microxanthone [12] except for the absence of the methoxy group at C-10. Thus **1** was determined to be 3,4,8-trihydroxy-6-methyl-9-oxo-9*H*-xanthene-1-carboxylic acid and trivially named carneusone A.

Compound **2** was a yellowish powder with a molecular formula C_16_H_12_O_7_ based on the HRESIMS peaks at *m*/*z* 317.0661 ([M + H]^+^, calcd. 317.0656) and 339.0481 ([M + Na]^+^, calcd. 339.0475). The NMR data of **2** showed great similarity with **1** except for the presence of an extra methoxy group signal at δ_H_ 3.91 (3H, H-12) and δ_C_ 60.8 (C-12), respectively (Table 1 and Table 2). One hydroxyl group at δ_H_ 12.43 (1H, br s, OH-8) indicated the formation of an intramolecular hydrogen bond with the carbonyl group. The HMBC correlations from H-2 to C-3, C-4, C-9a, C-10, and from OCH_3_-12 to C-4 suggested the methoxy group was attached at C-4 of compound **2** (Figure 2). All the data confirmed **2** as 3,8-dihydroxy-4-methoxy-6-methyl-9-oxo-9*H*-xanthene-1-carboxylic acid and it was trivially named carneusone B.

Compound **3** was isolated as a yellowish powder, and its molecular formula was determined to be C_16_H_14_O_8_ from the HRESIMS peak at *m*/*z* 357.0588 ([M + Na]^+^, calcd. 357.0581). The ^1^H NMR spectrum of **3** displayed the signals of three aromatic protons at δ_H_ 6.07 (2H, s, H-5 and H-7), 7.00 (1H, s, H-2), one methoxy group proton at δ_H_ 3.83 (3H, H-12), and one methyl proton at δ_H_ 2.15 (3H, s, H-11) (Table 1). The ^13^C NMR and HSQC spectra of **3** indicated the presence of one methyl group at δ_C_ 21.6 (C-11), three aromatic methines at δ_C_ 107.5 (C-5 and C-7), 104.1 (C-2), eleven quaternary carbons including nine aromatic carbons at δ_C_ 109.8 (C-8a), 119.4 (C-9a), 128.1 (C-1), 137.7 (C-4), 141.9 (C-4a), 146.7 (C-6), 146.8 (C-3), and 161.6 (C-4b and C-8), and two carbonyl carbons at δ_C_ 167.6 (C-11), 200.6 (C-9) (Table 2). The HMBC spectrum showed correlations from H-2 to C-1, C-3, C-4, C-9, and C-9a, from H-5/H-7 to C-4b, C-6, C-8, C-9, and C-11, from CH_3_-11 to C-5, C-6, and C-7, and from OCH_3_-12 to C-4 (Figure 2). The data were similar to benzophenone-derivative cytosporaphenone A [13], except for extra methoxy group signals at δ_H_ 3.83 (3H, H-12) and δ_C_ 55.9 (C-12). The HMBC correlations from OCH_3_-12 to C-3 indicated the methoxy group was attached at the C-3 of compound **3** (Figure 2). Thus, **3** was elucidated as 2-(2,6-dihydroxy-4-methylbenzoyl)-3,4-dihydroxy-5-methoxybenzoic acid and trivially named carneusone C.

Compound **4** had the same molecular formula, C_16_H_14_O_8,_ as **3** based on the HRESIMS peaks at *m*/*z* 335.0767 ([M + H]^+^, calcd. 335.0761), and 357.0593 ([M + Na]^+^, calcd. 357.0581). The NMR data of **4** also exhibited great similarity with **3** except the HMBC correlations were from OCH_3_-12 to C-4 instead of from OCH_3_-12 to C-3 as in **3** (Table 1 and Table 2). Thus, **4** was determined as 2-(2,6-dihydroxy-4-methylbenzoyl)-3,5-dihydroxy-4-methoxybenzoic acid and trivially named carneusone D.

Compound **5** yielded as a brown amorphous powder. The molecular formula was determined to be C_20_H_18_O_8_ by the HRESIMS peak at *m*/*z* 387.1085 ([M + H]^+^, calcd. 387.1074). The ^1^H NMR spectrum of **5** displayed two aromatic protons at δ_H_ 6.83 (1H, H-7), 6.91 (1H, H-4), and two oxy-methine protons at δ_H_ 4.52 (1H, H-11) and 4.57 (1H, H-12), and three methyl protons at δ_H_ 1.06 (3H, H-15), 1.38 (3H, H-14), and 2.41 (3H, H-16) (Table 3). One hydroxyl group signal was observed at δ_H_ 11.96 (1H, br s, OH-8). The ^13^C NMR and HSQC spectra signals of **5** implied the presence of three methyl groups at δ_C_ 18.5 (C-16), 23.1 (C-15), and 29.9 (C-14), two oxy-methines at δ_C_ 61.7 (C-11) and 87.1 (C-12), two aromatic methines at δ_C_ 107.4 (C-4) and 118.6 (C-7), and thirteen quaternary carbons at δ_C_ 78.4 (C-13), 97.2 (C-10), 107.4 (C-4), 107.8 (C-9a), 108.9 (C-8a), 123.5 (C-5), 130.2 (C-4a), 133.5 (C-2), 139.2 (C-4b), 149.0 (C-6), 151.5 (C-1), 153.7 (C-3), 160.4 (C-8), and 189.7 (C-9). The HMBC spectrum exposed correlations from H-4 to C-2, C-3, C-4a, C-9a, and C-10, from H-7 to C-5, C-6, C-8, C-8a, and C-16, from H-11 to C-4b, C-5, C-6, and C-12, from H-12 to C-5, C-10, C-11, C-13, and C-14, and from CH_3_-14/CH_3_-15 to C-12, C-13. All these data showed great similarity with compound N-1447D [14] except for the absence of methoxy group at C-3 and an extra hydroxyl group at C-2 (Table 3). The relative configurations of H-11 and H-12 were proposed as α-orientations on the basis of NOESY correlations CH_3_-15 with H-11 and H-12, which was consistent with prenylated anthranol harunganol F (Table 3) [15]. The result was confirmed by the quantum chemical calculation of NMR data (qccNMR). A molecular Merck force field (MMFF) conformational search of four configurations, (10*S*,11*S*,12*S*)-**5a**, (10*R*,11*R*,12*R*)-**5b**, (10*S*,11*R*,12*S*)-**5c**, and (10*R*,11*S*,12*R*)-**5d**, was performed with CONFLEX 8.5 (Conflex Corp., Tokyo, Japan). Each isomer only gave one energy minimum with a 10 kcal/mol energy window, which was optimized using the DFT method at a B3LYP/6-31G(d) level with the Gaussian16 program package (Gaussian Inc., Wallingford, CT, USA). Gauge-independent atomic orbital (GIAO) calculations of the optimized conformers were performed at mPW1PW91/6-311+G(d,p) level in a DMSO solution. The sum of (10*S*,11*S*,12*S*)-**5a** and (10*R*,11*R*,12*R*)-**5b** isomers have 100% DP4+ probability (Appendix A), which confirmed that H-11 and H-12 were on the same side of the tetrahydropyran moiety of **5** [16]. The most probable absolute configuration was further determined as (10*S*,11*S*,12*S*) based on the comparison of calculated ECD curves of (10*S*,11*S*,12*S*)-**5** with the experimental CD spectrum (Figure 3). 

Compound **6** was obtained as a brown amorphous powder with molecular formula C_20_H_18_O_7_ by the HRESIMS peaks at *m*/*z* 371.1140 ([M + H]^+^, calcd. 371.1125) and *m*/*z* 393.0958 ([M + Na]^+^, calcd. 393.0945). Compound **6** has very similar NMR data to **5** except there was a ^13^C NMR upfield shift from δ_C_ 61.7 (C-11) in **5** to 27.1 (C-11) in **6** (Table 3). The HSQC spectrum and HMBC correlations from H-11 to C-4b, C-5, C-12, and C-13, from H-12 to C-10, C-5, C-11, C-13, and C-14, and from OH-8 to C-7 and C-8, indicated the hydroxyl group at C-11 in **5** was missing in **6**. The absolute configuration was identified as (10*S*, 12*S*) based on the comparison of calculated ECD curves with the experimental CD spectrum (Figure 3).

The known compounds were elucidated as orcinol (**7**) [17], cordyol C (**8**) [18], aspergilol C (**9**) [19], calyxanthone (**10**) [20], 3,7-dihydroxy-1,9-dimethyldibenzofuran (**11**) [21], 2-(2’-hydroxypropyl)-5-methyl-7-hydroxychromone (**12**) [22], and evariquinone (**13**) [23] by comparing NMR data with those previously published in the literature.

### 2.3. Anti-Inflammatory Activity Test

All the compounds were determined for inhibition on NO secretion using LPS induced RAW 264.7 cells. Compounds **5**, **6**, and **8** exhibited a moderate effect with EC_50_ values of 34.6 ± 0.9, 20.2 ± 1.8, and 26.8 ± 1.7 μM, while **11** showed a potent effect with an EC_50_ value of 2.9 ± 0.1 μM. The positive control dexamethasone had an EC_50_ value of 2.9 ± 0.1 μM (Figure 4).

## 3. Materials and Methods

### 3.1. General Experimental Procedures

The optical rotation was analyzed using a InsMark IP-digi3 polarimeter (InsMark, Shanghai, China). Circular dichroism was measured with a JASCO J-1500 circular dichroism spectrophotometer (JASCO, Easton, PA, USA). ^1^H-NMR, ^13^C-NMR, and 2D NMR spectra were recorded on a Bruker Ascend 500 spectrometer (Bruker, Billerica, MA, USA) with TMS as a reference. High-resolution TOFESIMS was performed on a WATERS Xevo G2-S Qtof Quadrupole Time of Flight Mass Spectrometry (Waters, Milford, MA, USA). Analysis and semi-preparative reversed-phase HPLC were performed on a Shimadzu LC-2030 liquid chromatograph (Shimadzu, Kyoto, Japan) with YMC-Pack ODS-A column 250 × 10 mm i.d., S-5 µm × 12 nm. Column chromatography (CC) was performed on a silica gel (200–300 mesh, Jiangyou Silica Gel Co., Ltd., Yantai, China) or CHROMATOREX C18 silica (Fuji Silysia Chemical Ltd., Kozoji-cho, Kasugai Aichi, Japan).

### 3.2. Isolation and Species Identification of Marine Fungus

The fungal strain GXIMD00543 was isolated from a sponge tissue sample that was collected from the Weizhou islands coral reef, Beibu Gulf, in December 2019. The sponge was collected and identified as *Haliclona* sp. by Dr. Xin-Ming Liu, Institute of Marine Drugs, Guangxi University of Chinese Medicine (Nanning, China) (Appendix A). 

The strain was deposited in the Institute of Marine Drugs, Guangxi University of Chinese Medicine, Nanning, China. The fungus was identified by its morphological features and the sequence analysis of the internally transcribed spacer (ITS) region of the rRNA gene. The ITS sequence was amplified from the genome DNA via PCR with primers (ITS1: 5′TCCGTAGGTGAACCTGCGG3′ and ITS4: 5′TCCTCCGCTTATTGATATGC3′). ITS sequences (see Appendix A) were then uploaded to the National Center of Biotechnology Information (NCBI) for BLAST analysis (GenBank accession number: OR501447). 

### 3.3. Fungal Fermentation

The fungal strain was statically cultivated in one hundred and fifty 1000 mL Erlenmeyer flasks, each containing modified solid rice medium (80 g of rice, 0.4 g of yeast extract, 0.4 g of glucose, 3.6 g of artificial sea salt, and 120 mL of H_2_O) for 35 days at room temperature. Then, the fermented cultures were extracted with EtOAc three times and concentrated *in vacuo* to obtain extract (230 g). 

### 3.4. Extration and Isolation

The extract (230 g) was subjected to a silica gel column (1000 g) and eluted with CH_2_Cl_2_/MeOH (100:0–80:20, *v*/*v*) to yield 9 fractions (Fr. 1–9). 

Fraction 5 was separated using a silica gel column and eluted with CH_2_Cl_2_/EtOAc (98:2–50:50, *v*/*v*) to give 18 sub-fractions (s*Fr*. 5-1–5-18). s*Fr.* 5-8 was separated using an ODS silica gel column and eluted with CH_3_CN/H_2_O (20:80–80:20, *v*/*v*) to give 20 sub-fractions (s*Fr*. 5-8-1–5-8-20). s*Fr.* 5-8-12 was subjected to a semi-preparation of HPLC (42% CH_3_CN/H_2_O) at a flow rate of 3 mL/min to obtain **6** (12 mg, t_R_ 27.2 min) and **11** (6 mg, t_R_ 33.4 min). s*Fr.* 5-9 was separated using an *ODS* silica gel column and eluted with CH_3_CN/H_2_O (25:75–65:35, *v*/*v*) to give 22 sub-fractions (s*Fr*. 5-9-1–5-9-22). s*Fr.* 5-9-9 was subjected to a semi-preparation of HPLC (42% CH_3_CN/H_2_O) at a flow rate of 3 mL/min to obtain **2** (11 mg, t_R_ 36.3 min), **7** (28 mg, t_R_ 39.7 min), and **9** (13 mg, t_R_ 49.6 min). s*Fr.* 5-9-10 was subjected to Sephedex (LH-20) and a further semi-preparation of HPLC (28% CH_3_CN/H_2_O) at a flow rate of 3 mL/min to obtain **8** (14 mg, t_R_ 19.5 min). s*Fr.* 5-10 was separated using an ODS silica gel column and eluted with CH_3_CN/H_2_O (25:75–65:35, *v*/*v*) to give 21 sub-fractions (s*Fr*. 5-10-1–5-10-21). s*Fr.* 5-10-9 was subjected to a semi-preparation of HPLC (33% CH_3_CN/H_2_O) at a flow rate of 3 mL/min to obtain **12** (14 mg, t_R_ 22.9 min). s*Fr.* 5-10-11 was subjected to a semi-preparation of HPLC (34% CH_3_CN/H_2_O) at a flow rate of 3 mL/min to obtain **5** (16 mg, t_R_ 25.4 min). s*Fr.* 5-10-16 was recrystallized by CH_2_Cl_2_/MeOH to obtain **13** (28 mg).

Fraction 6 was separated by ODS silica gel column and eluted with CH_3_CN/H_2_O (20:80–80:20, *v*/*v*) to give 19 sub-fractions (s*Fr*. 6-1–6-19). s*Fr.* 6-8 was subjected to a semi-preparation of HPLC (24% CH_3_CN/H_2_O) at a flow rate of 3 mL/min to give 4 sub-fractions (s*Fr*. 6-8-1–6-8-4). s*Fr.* 6-8-1 was subjected to a semi-preparation of HPLC (17% CH_3_CN/H_2_O) at a flow rate of 3 mL/min to obtain **3** (15 mg, t_R_ 31.0 min) and **4** (26 mg, t_R_ 36.7 min). s*Fr.* 6-11 was subjected to a semi-preparation of HPLC (37% CH_3_CN/H_2_O) at a flow rate of 3 mL/min to obtain **1** (10 mg, t_R_ 12.4 min) and **10** (15 mg, t_R_ 27.6 min).

### 3.5. Spectroscopic and Spectrometric Data

Carneusone A (**1**): Yellowish powder; UV (MeOH): *λ*_max_ (log ε) 204 (3.18), 253 (3.44), 327 (2.98) nm; ^1^H and ^13^ C NMR data, see Table 1 and Table 2; HRESIMS *m*/*z* 303.0510 ([M + H]^+^, calcd. 303.0499) and 325.0330 ([M + Na]^+^, calcd. 325.0319).

Carneusone B (**2**): Yellowish powder; UV (MeOH): *λ*_max_ (log ε) 205 (3.33), 244 (3.53), 312 (3.16) nm; ^1^H and ^13^ C NMR data, see Table 1 and Table 2; HRESIMS *m*/*z* 317.0661 ([M + H]^+^, calcd. 317.0656) and 339.0481 ([M + Na]^+^, calcd. 339.0475).

Carneusone C (**3**): Yellowish powder; UV (MeOH): *λ*_max_ (log ε) 214 (3.31), 277 (3.06) nm; ^1^H and ^13^ C NMR data, see Table 1 and Table 2; HRESIMS *m*/*z* 357.0588 ([M + Na]^+^, calcd. 357.0581).

Carneusone D (**4**): Yellowish powder; UV (MeOH): *λ*_max_ (log ε) 213 (3.28), 277 (2.90) nm; ^1^H and ^13^ C NMR data, see Table 1 and Table 2; HRESIMS *m*/*z* 335.0767 ([M + H]^+^, calcd. 335.0761), and 357.0593 ([M + Na]^+^, calcd. 357.0581).

Carneusone E (**5**): Brown amorphous powder; [α]_D_^21^ = +8.5 (*c* 0.9, MeOH); UV (MeOH): *λ*_max_ (log ε) 202 (3.28), 261 (2.81), 280 (2.67), 296 (2.31), 368 (2.75) nm; CD (MeOH) *λ*_max_ (Δε): 211 (+0.39), 216 (−0.41), 234 (+0.89), 247 (+0.28), 253 (+0.52), 291 (−0.11), 305 (+0.35), 344 (+0.32) nm; ^1^H and ^13^ C NMR data, see Table 3; HRESIMS *m*/*z* 387.1085 ([M + H]^+^, calcd. 387.1074).

Carneusone F (**6**): Brown amorphous powder; [α]_D_^21^ = +35.5 (*c* 0.5, MeOH); UV (MeOH): *λ*_max_ (log ε) 200 (3.43), 261 (2.98), 280 (2.88), 294 (2.44), 374 (2.94) nm; CD (MeOH) *λ*_max_ (Δε): 210 (+1.79), 223 (−0.36), 255 (+0.52), 272 (+0.26), 282 (+0.37), 324 (−0.14) nm; ^1^H and ^13^ C NMR data, see Table 3; HRESIMS *m*/*z* 371.1140 ([M + H]^+^, calcd. 371.1125) and *m*/*z* 393.0958 ([M + Na]^+^, calcd. 393.0945).

### 3.6. Computational Methods

MMFF and DFT/TDDFT calculations were performed with CONFLEX 8.5 (Conflex Corp., Tokyo, Japan) and Gaussian16 program package (Gaussian Inc., Wallingford, CT, USA), respectively. The MMFF94s conformational search-generated low-energy conformers within a 10 kcal/mol energy window were subjected to further geometry optimization and frequency calculation using the B3LYP/6-31G(d) method. The single-point energy of the optimized conformers was recalculated at the M06-2X/def2TZVP level. Thus, the Gibbs free energy, obtained by the sum of the single-point energy and the thermal correction was used for the relative thermal free energy (DG) calculation and following Boltzmann population analysis at 298.15 K. The TDDFT-calculated conformers were performed using the cam-B3LYP functional with def2TZVP basis set. The number of excited states per each molecule was set to 20. The CD spectra were generated by the program SpecDis V1.71 [24] using a Gaussian band shape from dipole-length dipolar and rotational strengths. The calculated spectra were finally generated from the Boltzmann weighting of each conformer. The Grimme’s dispersion (D3 version) was used for empirical dispersion correction. Solvent effects (in MeOH) were taken into account by using the default SCRF method integral equation formalism variant (IEFPCM) for the whole calculation.

The B3LYP/6-31G(d) optimized geometries of **5** were adopted for further NMR computation. GIAO calculations of the ^1^H and ^13^C NMR chemical shifts were accomplished by DFT at the mPW1PW91/6-311+G(d,p) level in DMSO solution. The calculated NMR spectroscopic data were averaged according to the Boltzmann distribution by the program Multiwfn 3.7 [25]. 

### 3.7. Anti-Inflammatory Activity Test

The inhibition of the NO production assay was performed according to the reported procedures [26]. The mouse macrophage RAW264.7 cell lines used in this study were purchased from the American Type Culture Collection (ATCC, Manassas, VA, USA) and maintained in Dulbecco’s Modified Eagle Medium (DMEM) containing 10% fetal bovine serum (FBS), 100 ug/L of streptomycin, and 100 IU/mL of penicillin at 37 °C and 5% CO_2_ atmosphere (PHCbi, Minato-ku, Tokyo, Japan). The viability of the RAW264.7 cells was determined by MTT assay. The RAW264.7 cells were seeded at a density of 5 × 10^5^ cells/well in 96-well plates and incubated for 24 h at 37 °C and 5% CO_2_. Then, the media in each well was replaced using fresh FBS-free DMEM media. Different concentrations of compounds (2.5, 5, 10, 20, 40 μM) were prepared in FBS-free DMEM to give a total volume of 100 µL in each well of a microtiter plate. After 1 h of treatment, the cells were stimulated with 1 µg/mL of LPS for 24 h. The presence of nitrite was determined in the cell culture media using a commercial Griess reagent kit (Thermo Fisher Scientific, Waltham, MA, USA). Protocols supplied with assay kit were used for the application of the assay procedure. Briefly, 100 µL of cell culture medium with an equal volume of Griess reagent in a 96-well plate was incubated at room temperature for 10 min. Then, the absorbance was measured at 540 nm in a microplate reader (PerkinElmer, Waltham, MA, USA). The amount of nitrite in the media was calculated from the standard curve of sodium nitrite (NaNO_2_). Dexamethasone (DXM) was used as a positive control.

## 4. Conclusions

In summary, six benzophenone derivatives (**1**–**6**) and seven known compounds were obtained from sponge-derived fungus *Aspergillus carneus* GXIMD00543. Compounds **5**, **6**, **8**, and **11** exhibited anti-inflammatory effects by inhibiting NO secretion using LPS induced RAW 264.7 cells.

## Figures and Tables

**Figure 1 marinedrugs-22-00063-f001:**
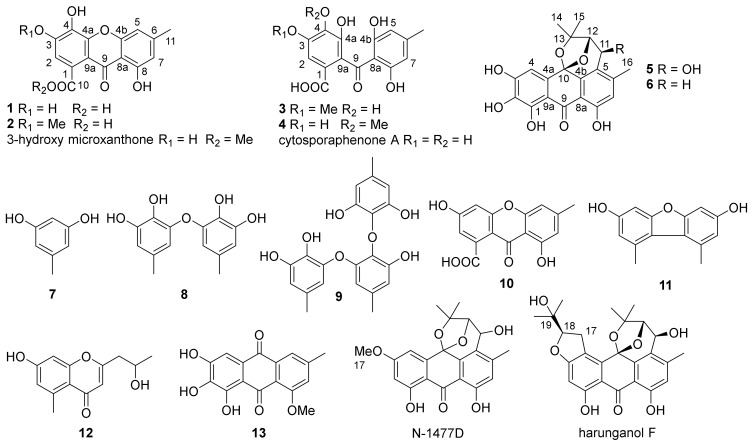
The chemical structures of compounds **1**–**13**, 3-hydroxy microxanthone [9], cytosporaphenone A [10], N-1477D [11], and harunganol F [12].

**Figure 2 marinedrugs-22-00063-f002:**
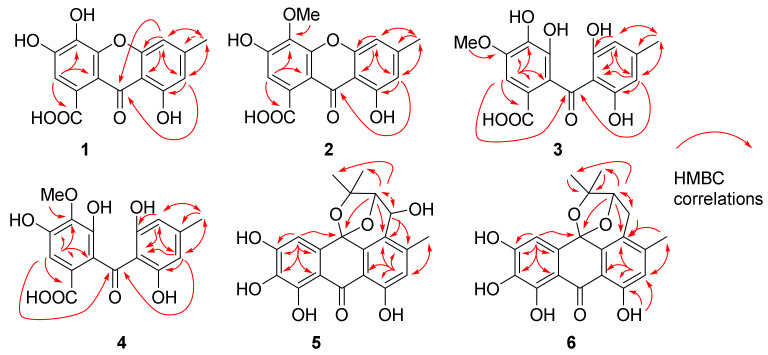
The key HMBC correlations of **1**–**6**.

**Figure 3 marinedrugs-22-00063-f003:**
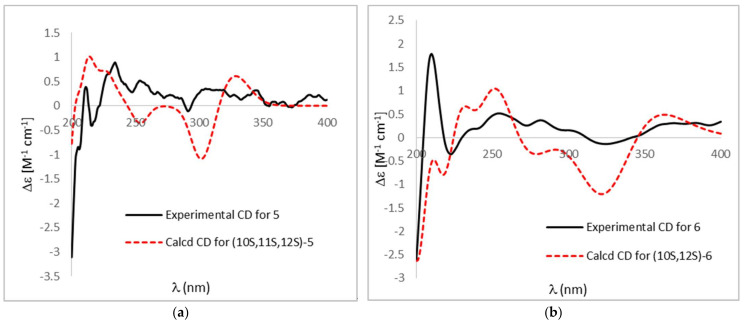
(**a**) Comparison of calculated ECD spectra of (10*S*,11*S*,12*S*)-**5** (red) in MeOH and experimental CD (black). σ = 0.16 eV, UV shift = 0 nm; (**b**) comparison of calculated ECD spectra of (10*S*, 12*S*)-**6** (red) in MeOH and experimental CD (black). σ = 0.30 eV, UV shift = 20 nm.

**Figure 4 marinedrugs-22-00063-f004:**
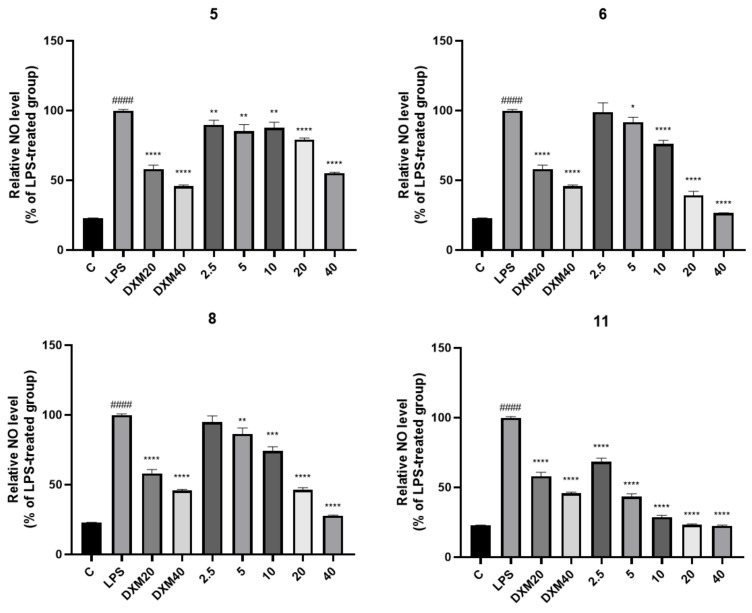
The inhibition of NO secretion in the RAW 264.7 cell line by the compounds **5**, **6**, **8**, and **11**. The inhibitory rate of NO in RAW 264.7 cells with different concentrations of compounds. Data were presented as mean ± SD of the experiments (n = 3). ^####^
*p* < 0.0001 compared with the blank control. * *p* < 0.05, ** *p* < 0.01, *** *p* < 0.001 and **** *p* < 0.0001 compared with the LPS model group. No cytotoxicity of compounds on RAW 264.7 cells was observed at 40 μM.

**Table 1 marinedrugs-22-00063-t001:** ^1^H NMR data for compounds **1**–**4**, 3-hydroxy microxanthone and cytosporaphenone A (*J* in Hz, *δ* in ppm).

No.	1 ^a^	2 ^a^	3 ^a^	4 ^a^	3-Hydroxy Microxanthone ^b^	Cytosporaphenone A ^c^
2	6.81, s	6.87, s	7.00, s	6.93, s	6.90, s	7.18, s
5	6.87, s	6.95, s	6.07, s	6.07, s	6.90, s	6.18, s
7	6.61, s	6.65, s	6.07, s	6.07, s	6.64, s	6.18, s
11	2.39, s	2.39, s	2.15, s	2.14, s	2.40, s	2.19, s
12		3.91, s	3.83, s	3.74, s	3.82, s	
OH-8	12.57, br s	12.43, br s				

^a^ Measured at 500 MHz in DMSO-*d*_6_; ^b^ measured in DMSO-*d*_6_ [12]; ^c^ measured in CD_3_OD [13].

**Table 2 marinedrugs-22-00063-t002:** ^13^C NMR data for compounds **1**–**4**, 3-hydroxy microxanthone and cytosporaphenone A (*J* in Hz, *δ* in ppm).

	1 ^a^	2 ^a^	3 ^a^	4 ^a^	3-Hydroxy Microxanthone ^b^	Cytosporaphenone A ^c^
No.	*δ*_C,_ Type	*δ*_C,_ Type	*δ*_C,_ Type	*δ*_C,_ Type	*δ*_C,_ Type	*δ*_C,_ Type
1	126.6, C	131.4, C	128.1, C	126.0, C	123.8, C	127.1, C
2	111.9, CH	112.5, CH	104.1, CH	108.6, CH	112.1, CH	109.8, CH
3	145.9, C	156.9, C	146.8, C	149.5, C	146.0, C	144.2, C
4	133.3, C	134.7, C	137.7, C	138.6, C	133.8, C	137.3, C
4a	152.3, C	150.6, C	141.9, C	146.6, C	151.3, C	142.1, C
4b	155.3, C	155.1, C	161.6, C	161.7, C	155.3, C	162.1, C
5	107.3, CH	107.5, CH	107.5, CH	107.5, CH	107.5, CH	107.9, CH
6	148.4, C	148.6, C	146.7, C	146.8, C	148.7, C	147.1, C
7	110.9, CH	111.2, CH	107.5, CH	107.5, CH	110.0, CH	107.9, CH
8	160.7, C	160.6, C	161.6, C	161.7, C	160.5, C	162.1, C
8a	105.6, C	105.6, C	109.8, C	109.6, C	105.6, C	109.1, C
9	179.8, C	179.4, C	200.6, C	200.2, C	179.7, C	200.0, C
9a	109.1, C	109.2, C	119.4, C	123.7, C	110.0, C	118.4, C
10	170.1, C	169.5, C	167.6, C	167.1, C	168.8, C	166.5, C
11	22.0, CH_3_	21.9, CH_3_	21.6, CH_3_	21.6, CH_3_	22.0, CH_3_	21.0, CH_3_
12		60.8, CH_3_	55.9, CH_3_	60.0, CH_3_	52.5, CH_3_	

^a^ Measured at 125 MHz in DMSO-*d*_6_; ^b^ measured in DMSO-*d*_6_ [12]; ^c^ measured in CD_3_OD [13].

**Table 3 marinedrugs-22-00063-t003:** ^1^H and ^13^C NMR data for compounds **5**, **6**, N-1447D and harunganol F (*J* in Hz, *δ* in ppm).

	5 ^a^		6 ^a^		N-1447D ^b^		Harunganol F ^c^	
No.	*δ*_C,_ Type	*δ* _H_	*δ*_C,_ Type	*δ* _H_	*δ*_C,_ Type	*δ* _H_	*δ*_C,_ Type	*δ* _H_
1	151.5, C		151.6, C		102.2, 106.9, 109.1, 109.7, 119.9, 122.2, 138.9, 140.2, 148.8, 162.1, 165.5, 166.9, 190.0, not assigned		166.3, C	
2	133.5, C		133.5, C				99.5, CH	6.36, s
3	153.7, C		153.5, C				168.0, C	
4	107.4, CH	6.91, s	107.4, CH	6.92, s		6.97, s	119.7, C	
4a	130.2, C		130.5, C				132.9, C	
4b	139.2, C		139.3, C					
5	123.5, C		121.0, C				121.9, C	
6	149.0, C		147.1, C				148.2, C	
7	118.6, CH	6.83, s	117.9, CH	6.80, s		6.81, s	120.0, CH	6.77, s
8	160.4, C		159.2, C				162.0, C	
8a	108.9, C		109.4, C				109.8, C	
9	189.7, C		189.8, C				189.6, C	
9a	107.8, C		108.1, C				108.9, C	
10	97.2, C		97.2, C		98.1, C		99.4, C	
11	61.7, CH	4.52, s	27.1, CH_2_	2.78, d, (17.4)	63.9, CH	4.58, s	63.6, CH	4.48, br d (10.0)
				3.04, dd, (17.4, 5.4)				
12	87.1, CH	4.57, s	81.1, CH	4.75, d, (5.4)	87.9, CH	4.72, s	87.6, CH	4.67, overlapped
13	78.4, C		82.1, C		79.2, C		78.3, C	
14	29.9, CH_3_	1.38, s	29.6, CH_3_	1.37, s	30.2, CH_3_	1.50, s	29.9, CH_3_	1.46, s
15	23.1, CH_3_	1.06, s	23.6, CH_3_	1.15, s	23.2, CH_3_	1.17, s	23.4, CH_3_	1.09, s
16	18.5, CH_3_	2.41, s	19.1, CH_3_	2.24, s	18.9, CH_3_	2.47, s	18.7, CH_3_	2.40, s
17					55.9, CH_3_	3.90, s	29.8, CH_2_	3.50, dd, (7.2, 16.7)
								3.37 dd, (9.5, 16.7)
18							91.3, CH	4.71, dd, (7.2, 9.5)
19							71.9, C	
20							24.5, CH_3_	1.23, s
21							25.7, CH_3_	1.35, s
OH-1				12.05, br s				
OH-8		11.96, br s		11.68, br s				

^a^ Measured in DMSO-*d*_6_; ^b^ Measured in CDCl_3_ [14]; ^c^ Measured in CDCl_3_ [15].

## Data Availability

The original data presented in the study are included in the article/Appendix A; further inquiries can be directed to the corresponding author.

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
