# Peer review of "Carneusones A-F, Benzophenone Derivatives from Sponge-Derived Fungus Aspergillus carneus GXIMD00543"

_marinedrugs, 2024, doi:10.3390/md22020063_

Round 1

Reviewer 1 Report

Comments and Suggestions for Authors

In the abstract “Their chemical structures were,,,,, quantum chemical calculation”: That means the authors applied calculations for plural compounds. However, the XYZ data for the compounds 5a and 6a were shown in SI. (“quantum chemical calculation” should be “quantum chemical calculations”)

The raw results and the detailed analysis of chemical shift calculations were found anywhere.

The reviewer disagrees with that calculated ECD spectra showed accordance with those of experimental. There are considerable discordances, Only the signs of the biggest one were identical.  

The exact mass of C15H11O7+ is 303.4993. The authors need to remove one electron.

In SI, there are so many Chinese characters. 

Author Response

Comment 1. In the abstract “Their chemical structures were,,,,, quantum chemical calculation”: That means the authors applied calculations for plural compounds. However, the XYZ data for the compounds 5a and 6a were shown in SI. (“quantum chemical calculation” should be “quantum chemical calculations”)

Response: The description “quantum chemical calculation” was revised as “quantum chemical calculations”. The 13C and 1H NMR chemical shift calculations for the relative configuration determination of compound 5 were supplemented (Figrue S36). The experimental CD of compounds 5 and 6 were re-determined. Also we calculated ECD of 5 and 6 again and supplied the comparisons with experimental CD curves in the revised manuscript (Figure 3).

Comment 2. The raw results and the detailed analysis of chemical shift calculations were found anywhere.

The reviewer disagrees with that calculated ECD spectra showed accordance with those of experimental. There are considerable discordances, Only the signs of the biggest one were identical.  

Response: The 13C and 1H NMR chemical shift calculations for the relative configuration determination of compound 5 were supplemented and analyzed (Figrue S36). Also the experimental CD of 5 and 6 were determined again and compared with calculated CD in the revised manuscript (Figure 3).

Comment 3. The exact mass of C15H11O7+ is 303.4993. The authors need to remove one electron.

Response: The HRESIMS data of compound 1 was revised as “m/z 303.0510 ([M + H]+, calcd. 303.0499) and 325.0330 ([M + Na]+, calcd. 325.0319)”.

Comment 4. In SI, there are so many Chinese characters. 

Response: The Supporting Information materials were carefully revised.

Reviewer 2 Report

Comments and Suggestions for Authors

The authors presented the structures and biological activities of six new benzophenone derivatives (1 – 6) from a strain of sponge-derived marine fungus Aspergillus carneus GXIMD 00543. However, this manuscript is not recommended for publication in its present form, but may be reconsidered as a new paper after throughout revisions. Listed below are my specific comments.

1.    Line 65; The structural formula and NMR data of 3-hydroxy microxanthone should be added to the manuscript for comparison with compound 1.

2.    Line 93; The structural formula and NMR data of cytosporaphenone A should be added to the manuscript for comparison with compound 3.

3.    Line 110; The compound number is "5" instead of "3". It should be corrected.

4.    Line 121; The structural formula and NMR data of compound N-1447D should be added to the manuscript for comparison with compound 5.

5.    Line 124; The structural formula and NMR data of harunganol F should be added to the manuscript for comparison with compound 5.

6.    Line 105, Table 2; In the 1H-NMR data of compound 6, the 4.75 ppm signal assigned to H-12 should be corrected to "d, 5.4" instead of "dd, 5.4, 11.0" from Figure S28.

7.    Lines 125 and 134, Figures S36 and S38; The observed ECD spectra of compounds 5 and 6 are not consistent with their calculated spectra. Also, which chromophore in their structures does the Cotton effect observed in the ECD spectra of compounds 5 and 6 originate from? The authors should comment on the above points.

8.    Figures S2 and S7; The signals at 12.57 and 12.43 ppm observed in the 1H-NMR spectra of compounds 1 and 2, respectively, are considered to be hydrogen of the hydroxyl group at the C-8. This is because an intramolecular hydrogen bond is formed between the OH-8 and the carbonyl group at C-9. The authors should comment on the above signals in the manuscript.

9.    Figures S12 and S17; The authors should comment in the manuscript on the possible intramolecular hydrogen bonding between the OH-8 and the carbonyl group at C-9 observed in the 1H-NMR spectra of compounds 3 and 4, respectively.

10.  Figure S22; The authors should comment on the signal at 11.96 ppm observed in the 1H-NMR spectrum of compound 5.

11.  Figure S28; The signals at 11.68 and 12.05 ppm observed in the 1H-NMR spectrum of compound 6 are considered to be hydrogen of the hydroxyl groups at the C-1 and C-8. This is because an intramolecular hydrogen bond is formed between the OH-1 and -8 and the carbonyl group at C-9. The authors should comment on the above signals in the manuscript.

12.  Supporting Information, p.23 – 28; Known compound numbers should match those in the text. Also, the descriptions of the above known compounds should be changed from Chinese to English.

Reviewer 3 Report

Comments and Suggestions for Authors

 Journal: Marine Drugs (ISSN 1660-3397)

Manuscript ID: marinedrugs-2826008

Type: Article

Title: Carneusones A-F, benzophenone derivatives from sponge-derived fungus Aspergillus carneus GXIMD00543

Compound 1

1)      Line 62: two carbonyl carbon at dC 170.1 (C-10), inserted of C-11

2)      Line 63: From H-7 to C-5, C-8, C-9: need to add the HMBC correlation from H-7 to C-9 in fig. 2. This is long range HMBC correlation can you check the HMBC correlation from H-7 to C-9 if found?

3)      You need to explain why we have a 4-pyrone, because no free proton (HBC Correlation) is found to attach the blocs A, B and C

4)      Line 64 and 65: All those data exhibited close similarity with compound 3-hydroxy microxanthone [9] except for the absence of methoxy group at C-10 instead of C-11

Compound 3

1)      Line 91: The HMBC spectrum showed correlations from H-2 to C-1, C-3, C-4, C-9, C-9a, C-11.

H-2 can not have the HMBC correlation with C-11, too fare.

Compound 5

1)      Line 110:  The 1H NMR spectrum of 5 displayed […], instead of: The 1H NMR spectrum of 3.

3.3. Fungal Fermentation

Line 179: […] to obtain extract (230 g) instead of (230g): Need a space.

Author Response

Comments and Suggestions for Authors

Compound 1

Comment 1)      Line 62: two carbonyl carbon at dC 170.1 (C-10), inserted of C-11

Response: The mistake was corrected in the new version.

Comment 2)      Line 63: From H-7 to C-5, C-8, C-9: need to add the HMBC correlation from H-7 to C-9 in fig. 2. This is long range HMBC correlation can you check the HMBC correlation from H-7 to C-9 if found?

Response: Long-range correlations (4JC,H) could be clearly found in HMBC spectra of compounds 1-5. Maybe it’s resulted from hyperconjugated effects of benzophenone derivatives.

Comment 3)      You need to explain why we have a 4-pyrone, because no free proton (HBC Correlation) is found to attach the blocs A, B and C

Response: The chemical structure elucidation of compound 1 was mainly depended on the NMR data comparison with known compound 3-hydroxy microxanthone. Also correlations from H-5 and H-7 to C-9 were clearly observed in HMBC spectrum (Figure S5). We have supplemented those two key correlations in Figure 2.

Comment 4)      Line 64 and 65: All those data exhibited close similarity with compound 3-hydroxy microxanthone [9] except for the absence of methoxy group at C-10 instead of C-11

Response: The mistake was corrected.

Compound 3

Comment 1)      Line 91: The HMBC spectrum showed correlations from H-2 to C-1, C-3, C-4, C-9, C-9a, C-11.

H-2 can not have the HMBC correlation with C-11, too fare.

Response: This error was corrected in the revision.

Compound 5

Comment 1)      Line 110:  The 1H NMR spectrum of 5 displayed […], instead of: The 1H NMR spectrum of 3.

Response: The mistake was corrected.

3.3. Fungal Fermentation

Comment 1) Line 179: […] to obtain extract (230 g) instead of (230g): Need a space.

Response: The mistake was corrected.

Reviewer 4 Report

Comments and Suggestions for Authors

In this manuscript, authors investigated the chemical constituents of the sponge-derived fungus Aspergillus carneus GXIMD00543, led to the discovery of 13 phenolic compounds. Among them, six were new benzophenone derivatives. Their structures were established by the analysis of NMR data and comparison with those reported in literature. Moreover, four isolates exhibited anti-inflammatory activities. The work was important, which could be attractive for readers.

However, minor revisions were required based on the following issues.

1.     It is better to introduce the reported secondary metabolites from the fungus Aspergillus carneus. Readers might want to know whether benzophenone derivatives had been reported from this species before.

2.     Please provide the retention times for the compounds purified by HPLC in the subsection ‘3.4. Extration and Isolation’.

3.     Please assign the broad singlet peaks or singlet peaks at δH 12.57, 12.43, 11.96, 12.05, 11.68 ppm in the 1H NMR spectra of compounds 1, 2, 5, and 6.

4.     Please provide the types of carbon signals in Table 1.

5.      Please check the coupling constants of H-12 of compound 6 in Table 2, which was not consistent with the observed in Figure S28.

6.     It is important to determine the relative configuration of C-10 before ECD calculations for compounds 5 and 6.

Other revisions:

1. Please add the word ‘of’ after the phrases ‘EC50 values’ or ‘EC50 values’ throughout the whole manuscript.

2. Figure 1 caption: ‘chemical structure’ → ‘chemical structures’

3. P2L11&P4L90: ‘two carbonyl carbon’ → ‘two carbonyl carbons’

4. P2L11: ‘170.1 (C-11)’ → ‘170.1 (C-10)’

5. P5L113: ‘three methyl group’ → ‘three methyl groups

6. Please provide the name of the sponge sample in the subsection ‘3.2. Isolation and Species Identification of Marine Fungus’.

7. Supporting Information P23–28: Please check the compounds and only keep those reported in the manuscript. And pay attention to the language.

Comments on the Quality of English Language

There were a few grammar or typo errors. Some of them were given in the comments to the authors.

Author Response

Comments and Suggestions for Authors

In this manuscript, authors investigated the chemical constituents of the sponge-derived fungus Aspergillus carneus GXIMD00543, led to the discovery of 13 phenolic compounds. Among them, six were new benzophenone derivatives. Their structures were established by the analysis of NMR data and comparison with those reported in literature. Moreover, four isolates exhibited anti-inflammatory activities. The work was important, which could be attractive for readers.

However, minor revisions were required based on the following issues.

Comment 1.     It is better to introduce the reported secondary metabolites from the fungus Aspergillus carneus. Readers might want to know whether benzophenone derivatives had been reported from this species before.

Response: We appreciate for your helpful suggestion. However we didn’t find benzophenone derivatives isolated from fungus Aspergillus carneus. In the revision we introduced several bioactive benzophenone derivatives reported from Aspergillus species.

Comment 2.     Please provide the retention times for the compounds purified by HPLC in the subsection ‘3.4. Extration and Isolation’.

Response: The retention times for the compounds purified by HPLC were supplemented in the revision.

Comment 3.     Please assign the broad singlet peaks or singlet peaks at δH 12.57, 12.43, 11.96, 12.05, 11.68 ppm in the 1H NMR spectra of compounds 125, and 6.

Response: The hydroxyl group signals observed in the 1H NMR spectra of compounds 1, 2, 5, and 6 were assigned in the Table 1 and Table 3.

Comment 4.     Please provide the types of carbon signals in Table 1.

Response: the types of carbon signals were supplemented in Table 1.

Comment 5.      Please check the coupling constants of H-12 of compound 6 in Table 2, which was not consistent with the observed in Figure S28.

Response: It was revised in the revision.

Comment 6.     It is important to determine the relative configuration of C-10 before ECD calculations for compounds 5 and 6.

Response: The relative configurations of compound 5 were elucidated by NOESY spectrum. Also we supplemented the 13C and 1H NMR chemical shift calculations to confirm the relative configuration assignment in the revision (Figrue S36).

Other revisions:

Comment 1. Please add the word ‘of’ after the phrases ‘EC50 values’ or ‘EC50 values’ throughout the whole manuscript.

Response: They were corrected in the revision.

Comment 2. Figure 1 caption: ‘chemical structure’ → ‘chemical structures’

Response: The mistake was revised.

Comment 3. P2L11&P4L90: ‘two carbonyl carbon’ → ‘two carbonyl carbons’

Response: The mistake was corrected in the revision.

Comment 4. P2L11: ‘170.1 (C-11)’ → ‘170.1 (C-10)’

Response: The mistake was revised.

Comment 5. P5L113: ‘three methyl group’ → ‘three methyl groups’

Response: It was revised in the new version.

Comment 6. Please provide the name of the sponge sample in the subsection ‘3.2. Isolation and Species Identification of Marine Fungus’.

Response: The sponge sample information was supplemented in the 3.2.section and Figure S1.

Comment 7. Supporting Information P23–28: Please check the compounds and only keep those reported in the manuscript. And pay attention to the language.

Response: The Supporting Information materials were carefully revised.

Comments on the Quality of English Language

Comment. There were a few grammar or typo errors. Some of them were given in the comments to the authors.

Response: The manuscript was carefully revised.

Round 2

Reviewer 2 Report

Comments and Suggestions for Authors

This revised manuscript has been modified according to the reviewer’s comments. It is acceptable for publication.

Reviewer 4 Report

Comments and Suggestions for Authors

Authors have addressed all my concerns. Only typo errors were found as following:

1. P1L45: ‘IC50 values range of’ → ‘IC50 values of’

2. P2L53: ‘MIC values range of’ → ‘MIC values of

3. ‘Table 1 and Table 2’ → ‘Tables 1 and 2

These could be revised for prof reading; thus, this manuscript can be accepted in the present form.